# Childhood Wheeze, Allergic Rhinitis, and Eczema in Hong Kong ISAAC Study from 1995 to 2015

**DOI:** 10.3390/ijerph192416503

**Published:** 2022-12-08

**Authors:** So-Lun Lee, Yu-Lung Lau, Wilfred Hing-Sang Wong, Lin-Wei Tian

**Affiliations:** 1Department of Paediatrics and Adolescent Medicine, Queen Mary Hospital, Hong Kong; 2Department of Paediatrics, Duchess of Kent Children’s Hospital, Hong Kong; 3Department of Paediatrics and Adolescent Medicine, The University of Hong Kong, Hong Kong; 4School of Public Health, The University of Hong Kong, Hong Kong

**Keywords:** ISAAC, wheeze, allergic rhinitis, eczema, parental atopy, school air quality

## Abstract

Background: The prevalence of allergic diseases has been increasing in developing areas but has reached a plateau in many developed areas. Regular surveys are imperative to assess the disease burden for the prioritization of resource allocation. Objectives: We examined the change in the prevalence of wheezing, allergic rhinitis (AR), and eczema in school-aged children with possible associative factors and possible health effects of school air quality. Methods: This was the third repeated cross-sectional study conducted in 2015–2016 using the International Study of Asthma and Allergies in Childhood (ISAAC) protocol. Our first and second surveys were conducted in 1994–1995 and 2000–2001, respectively. Regarding the third survey, we recruited 3698 children aged 6–7 from 33 local schools in 18 districts. Air quality, temperature, and humidity were also measured. The changes in prevalence, multiple regression, and GLIMMIX procedure were analyzed. Results: From our first survey to our third survey, the increased prevalences for lifetime wheeze, current wheeze, lifetime rhinitis, current rhinitis, current rhinoconjunctivitis, lifetime chronic rash, and current chronic rash were 4.2%, 2.1%, 12.5%, 12.6%, 14.2%, 3.9%, and 4.1%, respectively. Increased prevalence of parental atopy had the strongest association with an increased prevalence of each of these seven health outcomes. There was no significant association between school air pollutant levels and the prevalence of health outcomes. Conclusions: There was an increase in the prevalence of wheezing, allergic rhinitis, and eczema across the surveys. The most important associated risk factor identified was the increased prevalence of a parental history of atopy.

## 1. Key Messages

### 1.1. What Is the Key Question?

Has the prevalence of wheezing, allergic rhinitis (AR, and eczema condition been leveling off in developed areas as illustrated by Phase I (in the mid 1990s) and Phase III (in the early 2000) ISSAC2000s) ISAAC?

### 1.2. What Is the Bottom Line?

This is the first study to show a continual increase in wheeze, AR, and eczema prevalence using ISAAC methodology over 21 years in Hong Kong, a well-developed area.

### 1.3. Why Read on?

We identified significant associative factors for the observed increase in the prevalence that may cast light on preventive measures for these health outcomes that carry a significant global burden.

## 2. Introduction

Childhood allergic diseases are prevalent worldwide, posing a significant disease burden on patients, their families, and society [1,2,3,4,5]. The International Study of Asthma and Allergies in Childhood (ISAAC), the largest worldwide collaborative research project, showed that there was a wide global variation in the prevalence of asthma, allergic rhinitis (AR), and eczema with an increasing trend in developing countries and reaching a plateau in developed countries from the mid-1990s to the early 2000s [6]. Regular surveys are essential to assess the disease burden in terms of prevalence, economic impact, and effects on quality of life to prioritize resource allocation [7,8].

The first (1st) ISAAC survey was conducted in Hong Kong in 1994–1995, and showed that the prevalence of allergic diseases in school children aged 6 to 7 years old was comparable to that in Singapore and Great Britain [9]. The second (2nd) repeated cross-sectional survey in 2000–2001 showed a leveling off of lifetime asthma, lifetime wheeze, and current wheeze but a significant increase in the prevalence of current severe asthma symptoms, lifetime rhinitis, current rhinitis, and lifetime eczema [10]. Some environmental factors were associated significantly but not completely with the observable change. While viral infection is a well-known trigger for asthma exacerbation, local studies noted that it accounted for less than half of asthma exacerbation in clinics or hospital settings [11,12]. Instead, environmental air pollutants, including fine particles PM_2.5_ and coarse PM (PMC; 2.5–10 μm aerodynamic diameter), were confirmed to be significantly associated with hospital admission for childhood asthma [13] and emergency hospital admissions for respiratory diseases, respectively [14]. Most studies on the impact of environmental air pollutants on child health have focused on residential areas. A French study showed that poor air quality at school was related to an increased prevalence of allergic diseases [15]. In Hong Kong Special Administrative Region, China (HKSAR), around 20% of 650 primary schools are situated close to the main road, as defined by the Transport Department, at a mean distance of 20.5 m with a standard deviation (SD) of 24.7 m [16]. Yet there have not been any local studies on the health effect of school air pollution on school children. 

It has been more than a decade since our second ISAAC survey was conducted. We aimed to repeat the third (3rd). We also took this opportunity to study the effect of school air quality on the health outcomes assessed. The information may have potential implications for health care and urban planning and possible intervention strategies for disease prevention.

## 3. Aims and Objectives

We examined the change in the prevalence of wheezing, AR, and eczema in school children in HKSAR and explored possible associative demographic, family factors, respiratory tract infection (RTI), and other environmental factors. In addition, we also examined the possible association between school air quality by collecting indoor and outdoor air pollutants data with the prevalence of the seven primary health outcomes assessed, including lifetime wheeze, current wheeze, lifetime rhinitis, current rhinitis, current rhinoconjunctivitis, lifetime chronic rash, and current chronic rash.

## 4. Materials and Methods

### 4.1. Study Design

This was the 3rd of a series of repeated cross-sectional questionnaire surveys conducted in 2015–2016 involving a questionnaire survey and collecting indoor and outdoor air pollutants data at the schools when the questionnaires were distributed. The methodology was the same as the previous 2 surveys except for the slightly modified sampling frame. We recruited children from local schools located in 3 regions (Hong Kong Island (HK), Kowloon (KLB), and New Territories (NT)) at that time. Schools were selected from 18 refined districts within the aforementioned 3 regions geographically and administratively divided in HKSAR. Based on the modified population density, this approach gives a better representation of the child population in HKSAR and the additional measurement of air pollutants at school. The indoor environmental and health-related risk factors that affect the prevalence of wheezing, allergic rhinitis, and eczema were assessed by the core questionnaire, and the health outcomes were assessed by the environmental questionnaire provided by ISAAC [6,17,18] (see Appendix A). The study was approved by the Institutional Review Board of the University of Hong Kong/Hospital Authority Hong Kong West Cluster (HKU/HA HKWC IRB) (IRB ref. number: UW12-499).

### 4.2. Subjects

Same as our previous studies that recruited children aged 6 to 7 years [19]. We randomly approached each school until two schools from each district agreed to participate. Two classes with children aged 6 or 7 from each school were chosen. Children outside the specified age range were included in the data collection but excluded from the analysis. Parents were requested to fill in the ISAAC core and environmental questionnaire that included most of the potential demographic and indoor risk environmental factors reported in previous literature and 2 added new questions about traffic at residence [9,19].

### 4.3. School Air Quality Monitor

Air pollutants were measured indoors (in 2 classrooms) and outdoors from Monday to Friday (night and day) for two weeks when the questionnaire was distributed by the school to parents. Concentrations of particulate matter (PM_10_) were measured using filter-based samples collected by a PM_10_ monitor (TSI AM510) charged with a pump. The TSI AM510 allows measurement of PM_2.5_ or PM_10_ at one time only. A pilot run showed that the pump’s tubing was easily blocked when measuring PM_2.5_. Hence, PM_10_ measurement was chosen in order to obtain reliability. NASA data on the air quality in different areas of Hong Kong was provided by Hong Kong University of Science and Technology upon request. We used the data to estimate the air quality of PM_2.5_ at each school location [20]. Concentrations of nitrogen dioxide (NO_2_), ozone (O_3_), and sulfur dioxide (SO_2_) were measured by passive diffusion samplers using NO_2_ monitor (Z-1400XP), O_3_ monitor ZDL-1200 and SO_2_ monitor (Z-1300), respectively. Temperature and humidity data were also collected. During collection, the whole monitor was placed at a fixed position at one meter above ground level.

### 4.4. Estimation of Ambient Air Quality at Residential Locations

Geographical coordinates of the participants’ residential locations were identified from their residential addresses provided in the questionnaire. Surface extinction coefficients derived from NASA satellite data were used to estimate the residential ambient particulate air quality (PM_2.5_ and PM_10_) based on the residential locations [19]. The association of our outdoor air quality data with surface extinction coefficients (SEC) and aerosol optical depth (AOD) data from NASA satellites and air quality data from the nearest monitoring stations of the Hong Kong Environmental Protection Department (EPD) were determined. The satellite data and air quality data from EPD are freely available online. The information was used to adjust for the variation in exposure at home. 

### 4.5. Data Analysis

According to the instructions from ISAAC, missing answers on the questionnaire were merged with negative answers in the analysis. The Chi-square test was used to test the difference in the health outcomes (all categorical variables) related to wheeze, AR, and eczema between our 3rd survey and the 1st and 2nd surveys.

Trend analysis was conducted by Chi-square test across the 3 surveys. The trend in prevalence was considered significant only if both comparisons in 2015 vs. 2000 and 2015 vs. 1994 were significant with *p* < 0.025 and showed the trend in the same direction. Multi-test was adjusted by the Bonferroni correction. Multiple logistic regression was used to reveal any association between the change in the prevalence of potential risk factors and the change in the prevalence of 7 primary health outcomes. A total of 21 risk factors were included in the questionnaire [10]. The potential risk factors included in the regression analysis were those associated with a particular health outcome in univariate analysis (*p* < 0.05). They showed significant changes in prevalence between the study periods (*p* < 0.05). We also added an ordinal variable for the “study period” in the regression analysis and set our 1st survey as the reference period.

For our 3rd survey, the indoor and outdoor 10-day mean concentrations (microgram/m^3^) of each pollutant for each school were calculated. In addition, estimation of socioeconomic data (household size and individual income) was obtained from the census department in the tertiary planning unit (TPU) in Large Street Block Group (LSBG). A multi-level analysis model was employed. Firstly, a basic model was developed to detect any association between potential independent risk factors with 7 primary health outcomes using the Pearson chi-square test or exact Fisher test, followed by multiple logistic regression analysis.

The 2nd level of analysis involved 2 steps. First, between-school variability (“schools” variance) and between-class variability (“classes” variance) of the measured air pollutants with 7 primary health outcomes were estimated using generalized linear mixed models (SAS MIXED procedure) [15]. Then, significant factors identified in the basic model by multiple logistic regression, temperature and humidity data collected, and residential ambient particulate level derived from NASA satellite data was adjusted in the final analysis for the association between significant air pollutants and the 7 health outcomes by using generalized linear mixed models. All data were analyzed using SAS/PC (Cary, NC, USA) V9.1. A significance level of *p* < 0.05 was used for all analyses.

## 5. Results

The results of our first and second surveys were reported previously [10,11]. For our third survey, there were 3698 participants (51.4% male) aged 6 to 7 years old from 33 primary schools in HKSAR, compared to 3618 (50.9% male) in our first and 4448 (54.1% male) in our second survey.

The prevalences of wheezing, AR, and eczema in these three surveys were compared (Table 1). There had been a significant increase in the prevalences of all seven primary health outcomes. From our first survey to our third survey, the increased prevalences for lifetime wheeze and current wheeze, lifetime rhinitis, current rhinitis, current rhinoconjunctivitis, lifetime chronic rash, and current chronic rash were 4.2%, 2.1%, 12.5%, 12.6%, 14.2%, 3.9% and 4.1% respectively. There had also been a significant increase in the prevalence of secondary health outcomes, including the 12-month prevalence of nocturnal awakening with wheezing, nocturnal cough, and being kept awake by rash, and severe rhinitis interfering with daily activities.

Nine of twenty-one potential risk factors studied were found to be associated with the health outcomes in all three study periods and showed a significant prevalence change (Table 2).

All factors showed an increase in prevalence except for recurrent RTI in the last 12 months from the survey (≥4) (*p* < 0.01 for all).

The results of multiple logistic regression to examine if there is any association between significant prevalence change in potential risk factors, study period effect, and the increased prevalence of the seven primary health outcomes are tabulated in Table 3a–c, respectively. There were associations between the increased prevalence of parental history of atopy with the increased prevalence of each of the seven primary health outcomes; the increased prevalence of having small sibship size with the increased prevalence of lifetime wheeze, lifetime rhinitis, current rhinitis, and current rhinoconjunctivitis, and the increased prevalence of reporting frequent hospitalization for RTI in the 1st 12 months of life with the increased prevalence of lifetime wheeze and current wheeze. In addition, the increased prevalence of maternal smoking during the child’s 1st year of life was associated with lifetime wheeze and current wheeze, while that of current maternal smoking was associated with current rhinoconjunctivitis. The study period effect referred to potential environmental risks or protective factors that could not be addressed in the environmental questionnaire; for example, immigrants from mainland China, and the westernization of lifestyle. An increased prevalence of parental atopy had the strongest association with an increased prevalence of each of these seven health outcomes, and the odds ratio (OR) ranged from 2.35 to 3.12.

For the third survey, 11 risk factors were found to be significantly associated with the 7 primary health outcomes after multiple logistic regression. The results are shown in Table 4a,b.

Of the 33 schools that participated, 6 (18%) are situated close to the main road, with a distance of 20.5 + 24.7 m, compared to all schools of 51.1 + 40 m (*p* < 0.001). There was a significant association between the distance to the main road and school PM_10_ concentration, with a correlation r = 0.214 (*p* = 0.004). The school PM_10_ level collected by the monitor correlated reasonably well with the satellite PM_10_ level (*p* = 0.05). There were significant variations in the indoor (2 sites) and outdoor (1 site) air pollutant levels, including PM_10_, NO_2_, O_3_, and SO_2,_ at all the schools, and there was also significant variation across the schools. The overall mean concentrations of PM_10_, NO_2_, O_3_, and SO_2_ of the three sites collected over the 10 days for each school are depicted in Figure 1.

GLIMMIX multi-regression showed a significant association between school PM_2.5_ and lifetime wheeze, lifetime rhinitis, current rhinitis, and current rhinoconjunctivitis, and school NO_2_ and lifetime rhinitis, current rhinitis, and current rhinoconjunctivitis. There were no significant associations of school PM_10_, O_3_, SO_2_, or home PM_2.5_ or PM_10_ with each of the seven primary health outcomes. However, the significant associations between school PM_2.5_ and NO_2_ and health outcomes were lost after adjustment for multi-pollutant effects, confounding variables (significant risk factors associated with air pollutants, shown in Table 4a,b), temperature, and humidity (Table 5).

## 6. Discussion

Childhood allergic diseases are a significant public health problem in all countries regardless of the level of development [10,16]. Phase I and Phase III ISAAC showed that there was an increase in the prevalence of asthma, AR, and eczema (indicated by wheeze, rhinitis, and itchy rash, respectively) in developing countries and a leveling off or decreasing trend in developed countries from the mid-1990s to the early 2000s [6,17,18]. It has been speculated that environmental and lifestyle factors that induce asthma symptoms in genetically susceptible individuals may have reached saturation level [21]. Surprisingly, our third survey showed that the prevalence of wheezing, rhinitis, and itchy rash rose again after an apparent leveling off between our first and second surveys [22,23]. The increase was especially remarkable for the prevalence of lifetime and current eczema and current rhinoconjunctivitis with almost a two-fold increase. There have been a few similar studies reported previously, but the results varied considerably, the methodology might not be consistent across the study periods, and the study spans were noticeably shorter than in our study [24,25,26,27]. A study in Poland involving four repeated cross-sectional questionnaires in children aged 7 to 10 years from 1993 to 2014 [28] showed a distinct rise in the prevalence of lifetime physician-diagnosed asthma but not for current wheeze. To our knowledge, we are the first center in developed areas to show a continual increased prevalence of wheezing, rhinitis, and itchy rash using the same ISAAC protocol and questionnaire over 21 years [6].

Our study showed a consistent increase in the prevalence of wheezing, rhinitis, and itchy rash, which are closely linked. In addition, there has been a concurrent increase in the prevalence of current symptoms suggestive of more severe diseases, including nocturnal awakenings by wheezing, nocturnal cough, and rhinitis interfering with daily activities, and being kept awake by a chronic rash. It has been suggested that diagnostic exchange may account for the increases in prevalence [29]. This was refuted in our study as the prevalence of doctor-diagnosed asthma was consistently lower than the prevalence of wheezing across the three surveys.

Since our first survey, HKSAR has experienced a series of important socioeconomic and political events. Those that might have had a direct health impact include the Bird Flu pandemic in 2001, the SARS outbreak in 2003, the implementation of comprehensive smoking-free legislation in 2007, the Flu pandemic in 2009, the zero quota policy that restricted Mainland Chinese birth in 2013, and updating Air Quality Objectives in 2014 with the implementation of a wide range of air quality improvement measures. There were more participants with small sibship sizes, which might be due to various factors, including family choice, marital postponement for women that shortens their childbearing exposure period, and economic or social circumstances. The prevalence of parents smoking during the child’s 1st year of life and parental smoking at the time of the survey increased between the first and second surveys, but it seemed to level off in our third survey, possibly due to the effect of comprehensive smoking-free legislation [30]. There was an increase in the number of participants who reported frequent severe early RTI that required hospitalization in the 1st year of life. While parents with fewer children tend to be more cautious nowadays when their young child is sick, a possible increase in the incidence of young children with an inherent predisposition to more severe infection could not be excluded. In contrast, there was a decrease in frequent RTI in the last 12 months of the survey, which may reflect the general increase in public awareness about health protection from infectious diseases after several infectious epidemics.

An increased prevalence of parental history of atopy had the strongest association with the increasing prevalence of all seven health outcomes suggesting the importance of genetic predisposition. The relationship between RTI and the development of allergic diseases is intriguing. Early-life infections, especially if severe, are associated with the development of atopic disease, especially for asthma [31,32,33]. On the other hand, Strachan’s original hygiene hypothesis proposed that a lower incidence of infection in early childhood could explain the rapid rise in allergic diseases, especially hay fever [31,34]. Our study matched both theories as we showed the increased prevalence of frequent hospitalization for RTI in the 1st 12 months was associated with the increased prevalence of lifetime wheeze and current wheeze, while the increased prevalence of small sibship size, a surrogate for reduced microbial exposure, was associated with the increased prevalence of lifetime wheeze and current wheeze. We also noted that the increased prevalence of maternal smoking during the 1st year of a child’s life was associated with an increasing prevalence of lifetime wheeze and current wheeze but not other health outcomes [35,36]. We found a study period effect associated with the prevalence change in all seven primary health outcomes with adjusted ORs ranging from 1.12 to 2.26. Our previous study and other local studies suggested that outdoor environmental pollutants have a significant impact on hospital admission for asthma and other respiratory and allergy-related diseases, as there was a rapid decline in the air quality in HKSAR associated with vibrant economic activities in the region and nearby cities including Shenzhen and Guangzhou from the 1990s [13,14]. In our third survey, we did not find a significant association between school air quality and any health outcomes after multiple logistic regression, suggesting that genetic factors (parental history of atopy), life events (RTI in last 12 months), and indoor environmental factors (maternal smoking) could be more important associative risk factors. Although 18% of the schools recruited were situated closer to the main road compared to the other schools, we failed to demonstrate an association between school air quality and any health outcomes. In USA and Canada studies, the investigators used 400 m and 75 m, respectively, to define close proximity to main roads when assessing the impacts of air pollution. All the schools included in our study would have been defined as having close proximity to main roads accordingly [37,38]. As such, our observation could be genuine due to the statutory ban against the idling of motor vehicle engines implemented in December 2011 and the updating of Air Quality Objectives in 2014 with subsequent air quality improvement measures.

### Strengths and Limitations

The potential biases of the cross-sectional surveys were minimized by strict adherence to the ISAAC protocol and questionnaire. The large sample size for each survey and the summated sample size, the consistency of the results across the three surveys, the concurrent association of risk factors with the seven primary health outcomes, the narrow confidence intervals of the significant ORs, and the magnitude of the ORs of significant association matched closely with previous studies supported the belief that our methodology was robust. Some schools declined as they were worried that significant additional manpower commitment was required. Some schools were worried that if they participated in the study, their students might accidentally cause damage to air quality monitors. Due to the limited study period, we eventually recruited only 33 schools, with only 1 instead of 2 schools from 3 districts. However, we reached a sample size that provided a sufficiently large power for the study. Questions were added in our second and third surveys that were not in the first; for example, the types of fuels for cooking. Thus, we could only assess these factors if they also changed in prevalence. Due to the limited budgets, only three sets of air quality monitors were purchased. Two were placed in the classrooms, but there were usually six to eight classes for the two school years of each school. The air pollutant levels were generalized to all the other classes within the same school, potentially affecting our results. Satellite data for air pollutants were available for particulate matter but not for gaseous pollutants. Thus, our study might underestimate the health effect of residential air pollutants. We originally planned to extend the study to include more schools, but this was eventually suspended due to the emergence of the COVID-19 pandemic with school closures when we explored further research funding support.

## 7. Conclusions

Our study shows that the prevalences of wheezing, allergic rhinitis or rhinoconjunctivitis, and eczema have increased across three surveys. The most important associated risk factor identified was the increased prevalence of a parental history of atopy. We did not find school air pollutant levels to be significantly associated with the prevalence of these outcomes.

## Figures and Tables

**Figure 1 ijerph-19-16503-f001:**
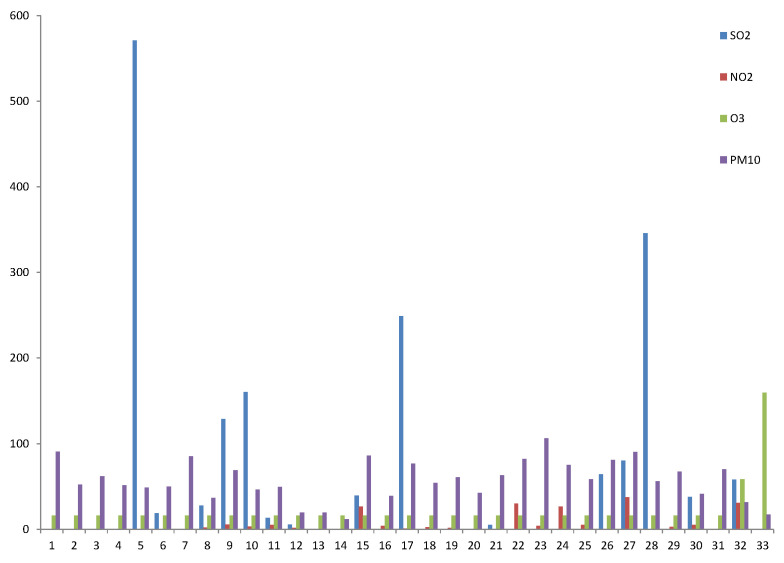
Overall mean concentrations of PM_10_, NO_2_, O_3_, and SO_2_ collected over the 10-day period for each school.

**Table 1 ijerph-19-16503-t001:** Self-reported prevalence of wheezing, rhinitis, and eczema.

		Prevalence in %			Prevalence Change across 3 Study Periods	*p*-Value	*p*-Value	Chi-Square Test for Trend **
		2015–2016	2000–2001	1994–1995				
Questions		*n* = 3698	*n* = 4448	*n* = 3618	*p*-Value	2015 vs. 2000	2015 vs. 1994	
Wheeze								
Wheeze ever (lifetime wheeze)	Yes	21.00	17.20	16.80	<0.01	<0.01	<0.01	↑
Wheeze in last 12 months (current wheeze)	Yes	11.30	9.40	9.20	<0.01	<0.01	<0.01	↑
Wheezing episodes in last 12 months	≥4	2.50	2.30	2.30	0.81	0.61	0.64	-
Night awakenings by wheeze in last 12 months *	>0	5.10	4.00	2.90	<0.01	0.01	<0.01	↑
Speech limitation during wheeze in last 12 months *	Yes	1.00	1.00	0.80	0.60	<0.01	0.44	-
Doctor diagnosed asthma ever	Yes	6.60	7.90	7.80	0.06	0.03	0.05	-
Exercise-induced wheeze in last 12 months *	Yes	6.00	7.70	7.10	0.01	<0.01	0.06	-
Nocturnal cough in past 12 months *	Yes	32.60	26.00	22.10	<0.01	<0.01	<0.01	↑
Allergic rhinitis								
Allergic rhinitis ever	Yes	35.30	33.90	31.90	<0.01	0.2	<0.01	-
Rhinitis ever (lifetime rhinitis)	Yes	51.40	42.40	38.90	<0.01	<0.01	<0.01	↑
Rhinitis in last 12 months (current rhinitis)	Yes	47.70	37.40	35.10	<0.01	<0.01	<0.01	↑
Rhinitis with itch eyes in last 12 months (current rhinoconjunctivitis)	Yes	27.80	17.20	13.60	<0.01	<0.01	<0.01	↑
Rhinitis interfering with daily activities *	M-S ^#^	7.50	4.80	2.10	<0.01	<0.01	<0.01	↑
Hay fever ever	Yes	1.00	1.40	1.20	0.27	0.13	0.50	-
Eczema								
Chronic rash ever (lifetime chronic rash)	Yes	9.60	5.40	5.70	<0.01	<0.01	<0.01	↑
Chronic rash in last 12 months (current chronic rash)	Yes	8.30	4.20	4.20	<0.01	<0.01	<0.01	↑
Chronic rash at typical areas	Yes	8.10	3.60	4.20	<0.01	<0.01	<0.01	↑
Age rash first occurred *	<2	46.20	31.80	29.20	<0.01	<0.01	<0.01	↑
Rash all cleared in last 12 months *	Yes	26.70	19.90	28.10	<0.01	<0.01	0.18	-
Kept awake by rash in last 12 months *	>0	4.90	3.00	1.60	<0.01	<0.01	<0.01	↑
Eczema ever	Yes	30.90	30.70	28.10	0.01	0.87	<0.01	-

* Subgroup with positive response to wheeze in last 12 months, rhinitis in last 12 months, and chronic rash in last 12 months. ^#^ Moderate to severe. ** The trends in prevalence were recognized if both *p*-values in 2015 vs 2000 and 2015 vs 1994 were less than 0.025. ↑ Increasing trend across the three study periods.

**Table 2 ijerph-19-16503-t002:** Prevalence of potential risk factors with significant change across 3 study periods.

	Prevalence in %			*p*-Value	*p*-Value		
	2015–2016 (*n* = 3698)	2000–2001 (*n* = 4448)	1994–1995 (*n* = 3618)	2015 vs. 2000	2015 vs. 1994	Prevalence Change across 3 Study Periods	Chi-Square Test for Trend **
Question	Yes	Yes	Yes			*p*-Value	
Born in Hong Kong	92.90	87.90	88.70	<0.01	<0.01	<0.01	↑
Siblings (≤2)	97.70	94.40	78.40	<0.01	<0.01	<0.01	↑
Hospital admission for RTI in 1st 12 months (≥4)	2.00	2.30	0.40	0.41	<0.01	<0.01	↑
RTI in last 12 months (≥4)	11.30	9.70	34.90	0.02	<0.01	<0.01	↓
Currently smoking mother	4.68	5.26	2.05	0.25	<0.01	<0.01	↑
Mother smoke during the child’s 1st 12 months of life	4.00	4.34	0.72	0.48	<0.01	<0.01	↑
Currently smoking father	28.23	30.04	21.45	0.08	<0.01	<0.01	↑
Father smoke during the child’s 1st 12 months of life	28.37	29.36	23.22	0.34	<0.01	<0.01	↑
Parental history of atopy	50.60	49.70	45.00	0.45	<0.01	<0.01	↑

RTI, respiratory tract infection; ↑ significant increase across 3 study periods, ↓ significant decrease across 3 study periods. ** The trends in prevalence were recognized if both *p*-values in 2015 vs. 2000 and 2015 vs 1994 were less than 0.025.

**Table 3 ijerph-19-16503-t003:** (**a**) Multiple logistic regression to assess association between risk factors and prevalence change in lifetime wheeze and current wheeze. (**b**) Multiple logistic regression to assess association of risk factors with prevalence change in lifetime rhinitis, current rhinitis, and current rhinoconjunctivitis. (**c**) Multiple logistic regression to assess association of risk factors with prevalence change in lifetime chronic rash and current chronic rash.

(**a**)						
		**Wheeze Ever (Lifetime Wheeze)**			**Wheeze in Last 12 Months (Current Wheeze)**	
		**Odds Ratio (95% CI)**	***p*-Value**		**Odds Ratio (95% CI)**	***p*-Value**
Siblings (≤2)		1.26 (1.04–1.52)	0.02		1.17 (0.92–1.49)	0.20
Hospitalization for RTI in 1st 12 months of age (≥4)		2.25 (1.64–3.08)	<0.01		2.36 (1.64–3.38)	<0.01
Currently smoking mother (Yes)		1.03 (0.76–1.38)	0.86		1.29 (1.13–1.90)	0.19
Mother smoke during the child’s 1st 12 months of life (Yes)		1.49 (1.08–2.06)	0.02		2.09 (1.41–3.10)	<0.01
Parental history of atopy (Yes)		2.69 (2.43–2.97)	<0.01		2.62 (2.29–2.99)	<0.01
Study period						
Study 1 (1994–1995)		Reference	Reference		Reference	Reference
Study 2 (2000–2001)		0.93 (0.82–1.06)	0.30		0.91 (0.77–1.06)	0.93
Study 3 (2015–2016)		1.21 (1.06–1.37)	<0.01		1.12 (0.95–1.31)	0.12
(**b**)						
	**Rhinitis Ever** **(Lifetime Rhinitis)**		**Rhinitis in Last 12 Months** **(Current Rhinitis)**		**Current Rhinoconjunctivitis**	
	**Odds Ratio (95% CI)**	***p*-Value**	**Odds Ratio (95% CI)**	***p*-Value**	**Odds Ratio (95% CI)**	***p*-Value**
Siblings (≤2)	1.19 (1.03–1.36)	0.02	1.30 (1.12–1.50)	<0.01	1.47 (1.20–1.81)	<0.01
Hospitalization for RTI in 1st 12 months of age (≥4)	1.30 (0.97–1.75)	0.08	1.27 (0.94–1.72)	0.12	1.15 (0.81–1.62)	0.44
Currently smoking mother (Yes)	1.00 (0.78–1.29)	0.98	1.04 (0.81–1.35)	0.74	1.54 (1.12–2.13)	0.01
Mother smoke during the child’s 1st 12 months of life (Yes)	1.17 (0.88–1.55)	0.29	1.15 (0.86–1.53)	0.34	1.21 (0.86–1.71)	0.28
Parental history of atopy (Yes)	2.39 (2.21–2.57)	<0.01	2.65 (2.46–2.87)	<0.01	3.12 (2.82–3.44)	<0.01
Study period						
Study 1 (1994–1995)	Reference	Reference	Reference	Reference	Reference	Reference
Study 2 (2000–2001)	1.19 (1.08–1.31)	0.01	1.08 (0.97–1.19)	1.20	1.20 (1.06–1.36)	<0.01
Study 3 (2015–2016)	1.73 (1.57–1.91)	<0.01	1.78 (1.60–1.97)	<0.01	2.26 (1.99–2.57)	<0.01
(**c**)						
		**Chronic Rash Ever** **(Lifetime Chronic Rash)**			**Chronic Rash in Last 12 Months** **(Current Chronic Rash)**	
		**Odds Ratio (95% CI)**	***p*-Value**		**Odds Ratio (95% CI)**	***p*-Value**
Siblings (≤2)		1.00 (0.76–1.33)	0.99		1.02 (0.75–1.39)	0.90
Hospitalization for RTI in 1st 12 months of age (≥4)		1.46 (0.90–2.37)	0.13		1.10 (0.60–1.99)	0.76
Currently smoking mother (Yes)		1.40 (0.86–2.30)	0.18		1.36 (0.78–2.37)	0.27
Mother smoke during the child’s 1st 12 months of life (Yes)		1.29 (0.77–2.16)	0.34		1.04 (0.57–1.89)	0.90
Parental history of atopy (Yes)		2.35 (2.02–2.75)	<0.01		2.35 (1.98–2.79)	<0.01
Study period						
Study 1 (1994–1995)		Reference	Reference		Reference	Reference
Study 2 (2000–2001)		0.94 (0.77–1.15)	<0.01		2.04 (0.72–1.12)	<0.01
Study 3 (2015–2016)		1.73 (1.43–2.09)	<0.01		1.83 (1.49–2.24)	<0.01

**Table 4 ijerph-19-16503-t004:** (**a**) Basic model for the associations between risk factors and 7 health outcomes in the 3rd survey 2015–2016 by multiple logistic regression. (**b**) Basic model for the associations between risk factors and 7 health outcomes in the 3rd survey 2015–2016 by multiple logistic regression

(**a**)								
	**Wheeze Ever** **(Lifetime Wheeze)**		**Wheeze in Last 12 Months** **(Current Wheeze)**		**Rhinitis Ever** **(Lifetime Rhinitis)**		**Rhinitis in Last 12 Months** **(Current Rhinitis)**	
	**Adjusted Odds Ratio** **(95% CI)**	**Adjusted** ***p*-Value**	**Adjusted Odds Ratio** **(95% CI)**	**Adjusted** ***p*-Value**	**Adjusted Odds Ratio** **(95% CI)**	**Adjusted** ***p*-Value**	**Adjusted Odds Ratio** **(95% CI)**	**Adjusted** ***p*-Value**
Sex (Male)	1.33 (0.63–0.88)	<0.01	1.33 (1.10–1.70)	<0.01	1.39 (1.30–1.59)	<0.01	1.38 (1.30–1.61)	<0.01
Hospital admission for RTI in 1st 12 months (≥4)	1.90 (1.14–3.15)	0.01	2.14 (1.22–3.74)	0.01	1.01 (0.62–1.64)	0.97	2.16 (1.71–2.71)	0.90
RTI in last 12 months (≥4)	1.77 (1.40–2.22)	<0.01	1.85 (1.42–2.41)	<0.01	2.00 (1.57–2.54)	<0.01	2.11 (1.66–2.68)	<0.01
Currently smoking mother (Yes)	1.11 (0.68–1.80)	0.68	1.18 (0.65–7.14)	0.58	1.02 (0.67–1.56)	0.93	1.12 (0.74–1.72)	0.59
Mother smoke during the child’s 1st 12 months of life (Yes)	1.82 (1.09–3.03)	0.02	2.09 (1.15–3.8)	0.01	1.29 (0.83–1.98)	0.26	1.43 (0.42–2.21)	0.11
Father smoke during the child’s 1st 12 months of life (Yes)	1.17 (0.98–1.41)	0.09	1.18 (0.94–1.49)	0.14	1.23 (0.96–1.31)	0.14	1.15 (0.99–1.34)	0.08
Parental history of atopy (Yes)	2.68 (2.25–3.18)	<0.01	2.27 (1.86–2.76)	<0.01	2.47 (2.14–2.84)	<0.01	2.63 (2.28–3.03)	<0.01
Nursery school (Yes)	1.31 (1.00–1.73)	0.05	1.47 (1.08–2.00)	0.01	1.17 (0.92–1.49)	0.20	1.13 (0.89–1.44)	0.29
Given paracetamol during baby period (0–2 years) (Yes)	1.17 (0.99–1.39)	0.07	1.22 (1.00–1.49)	0.04	1.40 (1.21–1.63)	<0.01	1.33 (1.14–1.54)	<0.01
Use town gas (Yes)	1.21 (1.01–1.45)	0.03	1.14 (0.92–1.91)	0.23	1.08 (0.94–1.24)	0.25	1.17 (1.02–1.35)	0.02
Use purifier in hot and cold seasons (≥1 per week)	1.23 (1.02–1.50)	0.03	1.18 (0.94–1.48)	0.10	1.28 (1.07–1.52)	0.01	1.22 (1.02–1.45)	<0.01
RTI, respiratory tract infection								
(**b**)								
	**Current Rhinoconjunctivitis**		**Chronic Rash Ever** **(Lifetime Chronic Rash)**		**Chronic Rash in Last 12 Months** **(Current Chronic Rash)**			
	**Adjusted Odds Ratio (95% CI)**	**Adjusted** ***p*-Value**	**Adjusted Odds Ratio (95% CI)**	**Adjusted** ***p*-Value**	**Adjusted Odds Ratio (95% CI)**			**Adjusted** ***p*-Value**
Sex (Male)	1.43 (1.25–1.67)	<0.01	1.14 (0.92–1.43)	0.24	1.25 (0.99–1.57)			0.05
Hospital admission for RTI in 1st 12 months (≥4)	1.05 (0.64–1.71)	0.84	1.30 (0.55–3.05)	0.55	1.42 (0.56–3.59)			0.46
RTI in last 12 months (≥4)	1.05 (0.64–1.71)	<0.01	1.64 (1.23–2.18)	<0.01	1.52 (1.13–2.03)			<0.01
Currently smoking mother (Yes)	1.05 (0.69–1.61)	0.83	1.19 (0.53–2.31)	0.78	1.16 (0.55–2.50)			0.69
Mother smoke during the child’s 1st 12 months of life (Yes)	1.45 (0.94–2.24)	0.09	1.05 (0.50–2.18)	0.89	1.06 (0.55–2.43)			0.69
Father smoke during the child’s 1st 12 months of life (Yes)	1.17 (1.00–1.37)	0.04	1.31 (1.00–2.18)	0.05	1.55 (1.17–2.07)			<0.01
Parental history of atopy (Yes)	2.66 (2.31–3.07)	<0.01	1.94 (1.58–2.37)	<0.01	1.98 (1.61–2.43)			<0.01
Nursery school (Yes)	1.15 (0.90–1.46)	0.26	1.02 (0.69–1.50)	0.9	1.00 (0.67–1.50)			0.98
Given paracetamol during baby period (0–2 years) (Yes)	1.33 (1.15–1.55)	<0.01	11.32 (1.08–1.62)	0.01	1.28 (1.00–1.61)			0.04
Use town gas (Yes)	1.18 (1.03–1.36)	0.02	11.05 (0.83–1.32)	0.69	1.29 (0.89–1.43)			0.33
Use purifier in hot and cold seasons (≥1 per week)	1.23 (1.44–1.46)	<0.01	11.11 (0.85–1.46)	0.43	1.04 (0.79–1.38)			0.73

**Table 5 ijerph-19-16503-t005:** Results of GLIMMIX multi-regression model for the associations between air pollutants and health outcomes.

		Wheeze Ever ^1^ (Lifetime Wheeze)			Rhinitis Ever ^2^ (Lifetime Rhinitis)			Rhinitis in Last 12 Months ^3^ (Current Rhinitis)		Current Rhinoonjunctivitis ^4^			
		Odds Ratio (95% CI)	*p*-Value	Adjusted *p*-Value *	Odds Ratio (95% CI)	*p*-Value	Adjusted *p*-Value *	Odds Ratio (95% CI)	*p*-Value	Adjusted *p*-Value *	Odds Ratio (95% CI)	*p*-value	Adjusted *p*-Value *
School	PM_2.5_	1.05 (1.01–1.10)	0.04	0.28	1.05 (1.00–1.10)	0.05	0.18	1.06 (1.01–1.11)	0.04	0.13	1.06 (1.01–1.11)	0.03	0.09
	PM_10_	1.00 (0.99–1.00)	0.33	-	1.00 (0.99–1.00)	0.95	-	1.00 (0.99–1.00)	0.89	-	1.00 (0.99–1.01)	0.87	-
	NO_2_	0.99 (0.98–1.01)	0.63	-	1.01 (1.01–1.03)	0.03	0.28	1.01 (1.01–1.03)	0.04	0.28	1.01 (1.01–1.02)	0.02	0.17
	SO_2_	1.00 (0.99–1.00)	0.74	-	1.00 (0.99–1.00)	0.92	-	0.99 (0.99–1.00)	0.99	-	0.99 (0.99–1.00)	0.91	-
	O_3_	0.99 (0.99–1.00)	0.31	-	0.99 (0.99–1.00)	0.35	-	0.99 (0.99–1.00)	0.54	-	0.99 (0.99–1.00)	0.47	-
Residence	PM_2.5_	1.00 (0.96–1.04)	0.88	-	0.99 (0.95–1.02)	0.48	-	0.99 (0.95–1.02)	0.48	-	0.99 (0.95–1.02)	0.50	-
	PM_10_	1.00 (0.97–1.03)	0.88	-	0.99 (0.97–1.02)	0.48	-	0.99 (0.97–1.02)	0.48	-	0.99 (0.97–1.02)	0.50	-

^1^ Wheeze ever (lifetime wheeze) values were adjusted with sex (male), hospital admission for RTI in 1st 12 months (≥4), RTI in last 12 months (≥4), mother smoke during the child’s 1st 12 months of life (Yes), parental history of atopy (Yes), use town gas (Yes), use purifier in hot and cold seasons (≥1 per week), and air pollutants by using multi-regression model. ^2^ Rhinitis ever (lifetime rhinitis) values were adjusted with sex (male), RTI in last 12 months (≥4), parental history of atopy (Yes), given paracetamol during baby period (0–2 years) (Yes), use purifier in hot and cold seasons (≥1 per week), and air pollutants by using multi-regression model. ^3^ Rhinitis values in last 12 months (current rhinitis) were adjusted with sex (male), RTI in last 12 months (≥4), parental history of atopy (Yes), given paracetamol during baby period (0–2 years) (Yes), use town gas (Yes), use purifier in hot and cold seasons (≥1 per week), and air pollutants by using multi-regression model. ^4^ Current rhinoconjunctivitis values were adjusted with sex (male), RTI in last 12 months (≥4), father smoke during the child’s 1st 12 months of life (Yes), parental history of atopy (Yes), given paracetamol during baby period (0–2 years) (Yes), use town gas (Yes), use purifier in hot and cold seasons (≥1 per week), and air pollutants by using multi-regression model. * A significance level of *p* < 0.05 was used for all analyses. All *p*-values were not significant after adjustment for the factors in Table 4, temperature and humidity. No significant results were observed for current wheeze, lifetime chronic rash, current chronic rash, and other pollutants were not significant.

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
