# Peer review of "Childhood Wheeze, Allergic Rhinitis, and Eczema in Hong Kong ISAAC Study from 1995 to 2015"

_ijerph, 2022, doi:10.3390/ijerph192416503_

Round 1

Reviewer 1 Report

The submitted manuscript is a comprehensive summary of the results of the third round of surveys in Hong Kong using the ISAAC protocol. It nicely integrates the previously found data to compare how the prevalence of allergic diseases changed over time. I especially like the discussion and that strengths and limitations are clearly stated. It is interesting to see how the prevalence of atopic diseases increases over such a long period.

My comments:

Materials and Methods: School air quality monitor: Please add the company names to the used measuring devices and a reference or URL for the source of the NASA satellite data.

Line 232: A “Figure 1” is mentioned, but I could not find any in the provided document. Please add this figure.

Discussion: I think that it would be good to discuss the study outcomes additionally by including the results of the surveys from the 13-14 year old children, where a decrease of wheeze (and the thereby expected asthma prevalence) has been reported. (Eur Respir J, 1997 Feb;10(2):354-60. doi: 10.1183/09031936.97.10020354 and Clin Exp Allergy, 2004 Oct;34(10):1550-5. doi: 10.1111/j.1365-2222.2004.02064.x) How could this phenomenon possibly be explained?

Author Response

Dear Editor & Reviewer,

I am writing to submit our revised manuscript “Childhood wheeze, allergic rhinitis and eczema in Hong Kong-ISAAC study from 1995 to 2015” for consideration at the International Journal of Environmental Research and Public Health. As the reviewers and editors suggested, we have performed English editing. The manuscript has then been revised according to the constructive comments from the reviewers and the editors as the attached documents have attached.

Reviewer 2 Report

This is an epidemiological study using the ISAAC methodology carried out in Hong Kong. The central point of the study is to add knowledge about prevalence of atopic diseases in childhood, as well as to increase the understanding of possible factors associated with changes in this prevalence. The paper has qualities to be published, but modifications and corrections are necessary for this.

First of all, the writing in English must be proofread by a native speaker. There are several sentences where the writing is truncated, making it difficult to understand the message, either because of the lack of adequate punctuation, or because of the construction of the sentence structure. 

Introduction:

Lines 48-52: "...showed that there was a wide global variation of the prevalences of asthma, allergic rhinitis (AR) and eczema with increasing trend in developing countries and reaching a plateau in developed countries from mid 1990s to early 2000[6]." Please, support this statement with other, preferable more recent, references.

Materials and Methods:

Line 99: "see Figure 2". Firstly, there is no previous reference to Figure 1, which makes the citation of a Figure 2 unjustifiable. Secondly, I could not analyze any figure, as they are not attached to the main document and are not available anywhere on the Journal´s reviewer area. 

Lines 99-101: "For the ISAAC questionnaire, the prevalences of wheeze, allergic rhinitis, eczema were assessed with questions on different indoor risk environmental factors and health-related risk factors [6, 17, 18] (see Appendix A)."  Example of a sentence that the reader needs to reread several times to understand its meaning. Rewrite please. 

Lines 106-113: "We recruited children aged 6 to 7 years old in the two previous surveys. We randomly approached each school until 2 schools from each district agreed to participate. Two school years from each school were chosen which included children with the greatest proportion of 6-year old and 7-year old. Children outside the specified age range were included in the data collection but excluded from analysis. Parents were requested to fill-in the ISAAC core questionnaire and environmental questionnaire that included most of the potential demographic and indoor risk environmental factors reported in previous literature, and 2 added new questions of traffic at residence."       Another example of a difficult paragraph to understand, either by English writing or by the fact that the authors start the sentence talking about "in the two previous surveys" (??)

Lines 122-123: "We used NASA satellite data to estimate the air quality and PM2.5 value at each school’s location."   Please, explain better how to obtain and use this NASA data to estimate air quality and PM2.5.

Lines 130-132: "Surface extinction coefficients derived from NASA satellite data were used to estimate the residential ambient particulate air quality (PM2.5 and PM10) based on the residential locations." Please, explain better, in the same way as in the previous item. 

Line 133: "...SEC and AOD data from NASA satellites and..."  The first time an abbreviation appears in the text (SEC / AOD), it has to be described and/or explained.

Lines 148-149: "21 potential risk factors were included in the questionnaire." Please, don´t start a sentence with numbers.   

Describe and reference the risk factors included in the ISAAC questionnaire, justifying the reasons for their inclusion. This justification is fundamental because such inclusion, in addition to modifying the original ISAAC questionnaire, made the questionnaire much more extensive and time-consuming to answer. 

Still about the questionnaire used, was it validated for use in the Hong Kong population? If yes, include this statement and reference it. 

Results:

Table 1: Review formatting. 

Lines 210-211: "There was also a study period effect, implying some unmeasured factors was associated with the observed change of all the 7 health outcomes."   Confused writing. Please, rewrite. 

Line 232: Here, Figure 1 is cited, while Figure 2 was cited in Line 99 (????). As previously described, I did not have access to both. 

Discussion:

Lines 257-260: "To our knowledge, we are the 1st centre in developed areas that showed a continual increase prevalences of wheezing, rhinitis and itchy rash using the same ISAAC protocol and questionnaire over a 21-year span." 

Please, reference this statement with recent studies that show no increase in the prevalence of atopic diseases in developed countries in the last 21 years. See note in Introduction (Lines 48-52).

Line 277: "...prevalences..." The terma prevalence is used in the plural form in several parts of the text. Please, check if this is the proper way to use it. 

Lines 293-298: "Our study matched both theories as we showed the increased prevalence of frequent hospitalization for RTI in 1st 12 months has associated to the increased prevalences of lifetime wheeze and current wheeze while the increased prevalence of small sibiship size, a surrogate for reduced microbial exposure has associated to the increase prevalences of lifetime wheeze, lifetime rhinitis, current rhinitis and current rhinoconjunctivitis." 

How do the authors explain this observation that distinct risk factors ("more infections versus less infections") point to the same end result (increased prevalence of wheeze, rhinits etc)?

Lines 301-305: "Our results were consistent with previous reviews that the relationship between exposure to second-hand smoking in early childhood and asthmatic symptoms and reduced lung function is strong, whereas the evidence related to the development of allergy is much weaker and the effect is higher for maternal as opposed to paternal smoking[34 - 35]."  Paragraph with unclear writing. Please, rewrite. 

Lines 318-319: "In USA and Canada studies, the investigators used with 500m and 75m respectively to define close proximately..."

Reference 36 refers to a study carried out in Mexico and not in the United States. 

...used with 500m and 75m respectively to define...  Unintelligible phrase. Rewrite and note that this type of problem recurs throughout the manuscript. Please, review the entire text, as previously stated. 

Lines 332-333: "Unexpectedly, we encountered difficulty in recruiting schools. Many schools declined because of the intensive approach and exhaustion." Explain better why many schools refuse to participate in the study, as this is not clear to the reader. 

References:

What do references 38, 39 and 40 refer to, if the last cited reference in the text is 37 (Line 321)?

Finally, in the Discussion or in Limitations section, the authors must explain why only after 7 years of data collection these results are being submitted for publication. 

Author Response

Dear Editor and reviewer,

I am writing to submit our revised manuscript “Childhood wheeze, allergic rhinitis and eczema in Hong Kong-ISAAC study from 1995 to 2015” for consideration at the International Journal of Environmental Research and Public Health. As the reviewers and editors suggested, we have performed English editing. The manuscript has then been revised according to the constructive comments from the reviewers and the editors as the attached cover letter below.

Round 2

Reviewer 2 Report

I will be concise in the observations, as this is a 2nd revision.

Writting in English still has problems, as we can see in the lines 25-26 (THERE WAS no significant association between school air pollutant levels AND the prevalence of the health outcomes), line 96 (Schools WERE selected from 18 refined districts...), line 236 (Of the 33 schools the (???) participated...), lines 308 and 310 (has associated or WAS associated?) and Line 321: allergically (???) diseases.

Lines 97-98: What is the purpose of bold in the words population density?

Line 100:  Figure 1 presents a result, and is not part of the Methods. Therefor, it should not be cited in this section.

Point 9: About the questionnaire used: my question about wheter the questionnaire was validated or not for use in Hong Kong population was not answered and/or referenced. 

Point 15: The corrected text is not inserted in the manuscript and the authors do not provide an explanation/hypothesis for this observation.

Point 20 (lines 344-350 in the revised manuscript): in the manuscript it is written in one way, in the comment on the side of the text another, and in the response to reviewer´s comments another again. Finally, what is the explanation for this 7 years delay in disclosing the data?

Author Response

Dear Editor and reviewer:

Following your advice, we have coordinately revised the relevant content and polished the English writing as well, what's more, accordingly to your suggestions in the last email:

a. We have checked all references.
b. 'Track change' is adopted in our word file.

A cover letter for this round is attached below.

Best Regards,

Dr. SL Lee
